# Body Stalk Anomaly

**DOI:** 10.3390/diagnostics14050518

**Published:** 2024-02-29

**Authors:** Nicolae Gică, Livia Mihaela Apostol, Iulia Huluță, Anca Maria Panaitescu, Ana Maria Vayna, Gheorghe Peltecu, Nicoleta Gana

**Affiliations:** 1Gynecology Department, Faculty of Medicine, Carol Davila University of Medicine and Pharmacy, 020021 Bucharest, Romania; gica.nicolae@umfcd.ro (N.G.); iuliahuluta16@gmail.com (I.H.); anca.panaitescu@umfcd.ro (A.M.P.); anamariavayna@gmail.com (A.M.V.); gheorghe.peltecu@umfcd.ro (G.P.); 2Clinical Hospital of Obstetrics and Gynecology Filantropia, 011171 Bucharest, Romania; livia-mihaela.cosma@rez.umfcd.ro

**Keywords:** body stalk anomaly, amniotic rupture, abdominal defect

## Abstract

Abdominal wall defects encompass three primary classifications: gastroschisis, omphalocele and anomalies resembling body stalk. Potential causative factors include early amnion rupture, amniotic bands, vascular disruptions or abnormal folding of the embryo. The prevalence of these defects stands at 1 in 14,000 live births. Body stalk anomaly is characterized by a substantial abdominal defect coupled with spine and limb anomalies, along with a very short or absent umbilical cord. We present a case of a rare abdominal defect known as body stalk anomaly, the most severe form of this spectrum of diseases. The diagnosis of this anomaly was established during the first trimester of pregnancy. Subsequently, the patient opted for pregnancy termination and chose not to undergo genetic testing. The anatomo-pathological results confirmed the findings. Body stalk anomaly is not compatible with life; therefore, early identification and understanding the clinical implications of this rare anomaly for informed decision-making in prenatal care are very important.

Body stalk anomaly (BSA) is a rare and serious syndrome characterized by severe malformations. Despite the ongoing research, the exact cause and mechanisms behind this condition remain unclear. However, certain risk factors have been identified, such as lower socioeconomic status, maternal substance abuse (particularly cocaine), maternal diabetes and maternal hemorrhagic disorders [1]. It is important to note that body stalk anomalies are typically not linked to chromosomal abnormalities, although genetic defects related to embryonic development may contribute to their occurrence [1,2,3,4,5,6]. The prevalence of this condition is 1 in 14,000 live births [2]. Body stalk anomaly is characterized by a large abdominal defect associated with spine and limb anomalies and a very short or absent umbilical cord [2]. There is no increased risk of recurrence in subsequent pregnancies. Body stalk anomaly is the most severe form of abdominal wall defect and is not compatible with life [3,5].

Ultrasound examination is an important diagnostic tool in the first trimester. The signs include abdominal wall defects; spine and limb abnormalities and, more specifically, abdominal contents herniated in the coelomic cavity; scoliosis and a very short umbilical cord [4]. For a better understanding, detailed information can be found in the diagram in Figure 1. 

A retrospective study of 17 cases of BSA diagnosed in a tertiary unit between 2009 and 2015 has proposed an algorithm for the differential diagnosis of abdominal wall defects in the first trimester [6]. This algorithm, based on specific diagnostic criteria, can aid in the accurate distinction between various abdominal wall defects, including BSA, during early pregnancy scans. The differential diagnoses of abdominal wall defects include exomphalos, gastroschisis, cloacal exstrophy and the OEIS complex, Pentalogy of Cantrell, abdominoschisis due to amniotic bands and BSA [6]. The abdominal organs being attached to the placenta, severe kyphoscoliosis and the absence of a free-floating umbilical cord are diagnostic of BSA. In exomphalos, the herniated viscera appear in the base of the umbilical cord, and a free-floating cord is visible in the amniotic cavity. In gastroschisis, Pentalogy of Cantrell and cloacal exstrophy, the eviscerated organs are within the amniotic cavity, and the umbilical cord is free-floating. In abdominoschisis due to amniotic bands, the amniotic membrane continuity is lost, but the umbilical cord is free-floating. In early amnion rupture, the deformation and disruption of other structures, including craniofacial and limbs, can be demonstrated (limb–body wall complex) [6].

On ultrasound examination, we found a fetus with a CRL of 52 mm corresponding to 11 weeks and 6 days and a nuchal translucency of 2 mm. There was a large abdominal wall defect with a membrane-covered mass outside the amniotic cavity containing the abdominal organs. 

We performed a transvaginal ultrasound to better visualize the defect. On the TV scan, we could demonstrate the upper part of the fetal body being inside the amniotic cavity and the internal organs, such as the liver and ductus venosus, the stomach and the bowels, being inside the celomic cavity (Figure 2a,b). Other ultrasound findings were a short umbilical cord and short and abnormally developed inferior lower limbs. We also noticed that normal views of the heart could not be obtained, suggesting double-inlet ventricle (Figure 3a), and there was a severe scoliosis of the fetal spine (Figure 3b).

The combined screening test for chromosomal abnormalities, performed using a combination of ultrasound markers, such as nuchal translucency, nasal bone, ductus venosus, tricuspid valve flow and biochemistry (b-HCG and PAPP-A), showed a high chance of aneuploidies. Invasive genetic testing was offered and declined by the patient. She was counselled on the poor outcome of these defects regardless of the genetic testing results, and she opted for the termination of pregnancy and no further genetic testing. She was admitted to hospital. The anatomo-pathological results confirmed the multiple abnormalities detected using ultrasound.

In conclusion, there is a high variability in the manifestations of this condition. Body stalk anomaly is a lethal condition that can be detected in the first trimester, and an early and accurate diagnosis is crucial to the appropriate counselling for and management of affected pregnancies.

## Figures and Tables

**Figure 1 diagnostics-14-00518-f001:**
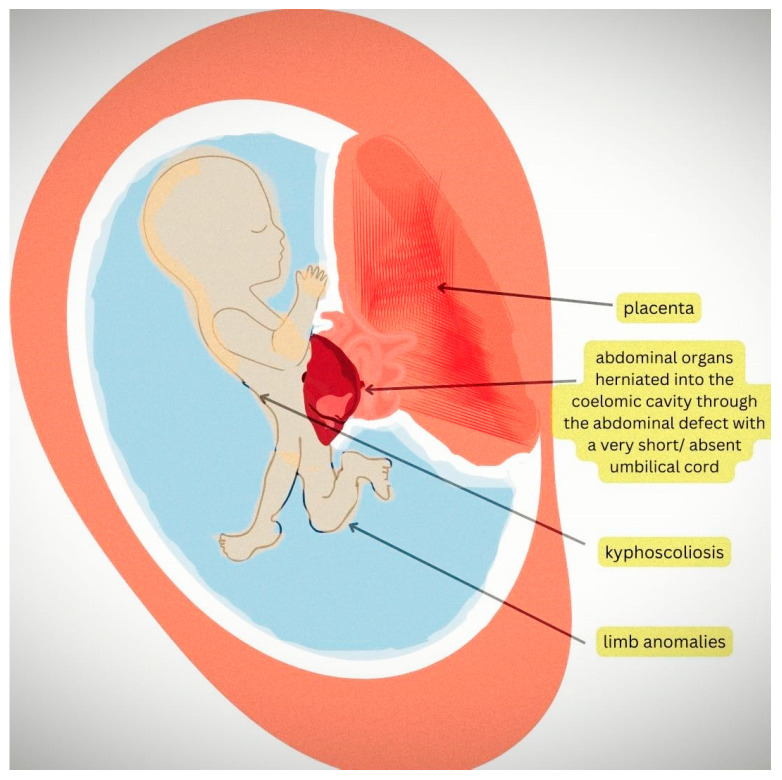
The diagram illustrates the ultrasound findings of a fetus with body stalk anomaly. In body stalk anomaly (BSA), the abdominal wall defect is a hallmark feature, and the abdominal organs, such as the liver and intestines, are typically herniated into the coelomic cavity due to this defect, with a very short or absent umbilical cord. Additionally, kyphoscoliosis (abnormal curvature of the spine) and limb abnormalities, such as shortened or absent limbs and clubfoot, are also visible in the image.

**Figure 2 diagnostics-14-00518-f002:**
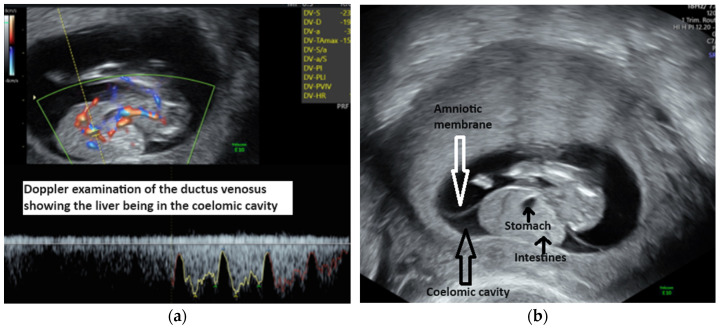
(**a**) Ductus venosus Doppler outside the abdominal cavity. (**b**) Visualization of the amniotic membrane and the abdominal organs inside the celomic cavity.

**Figure 3 diagnostics-14-00518-f003:**
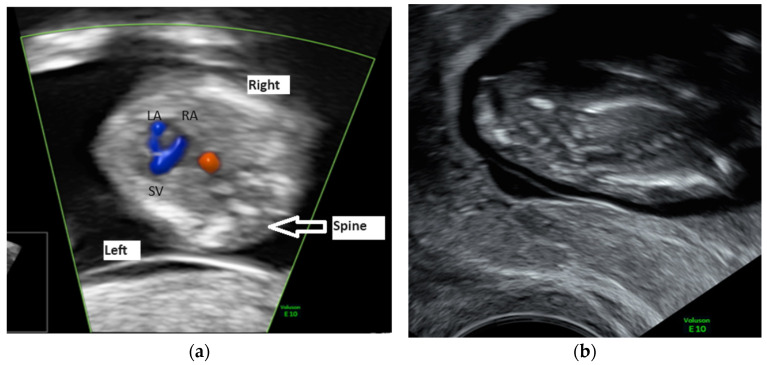
(**a**) The abnormal four-chamber view of the fetal heart suggesting double-inlet ventricle (LA—left atrium, RA—right atrium, SV—single ventricle). (**b**) Scoliosis of the fetal spine.

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
