# Peer review of "Body Stalk Anomaly"

_diagnostics, 2024, doi:10.3390/diagnostics14050518_

Round 1
Reviewer 1 Report
Comments and Suggestions for Authors
Thank you for providing the opportunity to review the paper titled "Body Stalk Anomaly". In this paper, the authors presented the ultrasound images of a fetus with body stalk anomaly. I would like to express the following concerns regarding this paper.
Figure 1(a) and 1(b)
The figures provided are too small to adequately show details. They should be enlarged, and the celomic cavity should be clearly indicated using arrows for better understanding.
Figure 2(a)
The abnormality in the four-chamber view is not discernible from the figure provided. It is recommended to enlarge the figure and include additional explanations for clarity.
Line 35
‘She had tree Caesarean Sections’ should be ‘She had three Caesarean Sections’.
References
Reference No. 4 appears to be a website and is not cited. I suggest omitting this reference.
Author Response
Dear Reviewer,
Thank you for providing the opportunity to review the paper titled "Body Stalk Anomaly". In this paper, the authors presented the ultrasound images of a fetus with body stalk anomaly. I would like to express the following concerns regarding this paper.
Figure 1(a) and 1(b)
The figures provided are too small to adequately show details. They should be enlarged, and the celomic cavity should be clearly indicated using arrows for better understanding.
Figure 2(a)
The abnormality in the four-chamber view is not discernible from the figure provided. It is recommended to enlarge the figure and include additional explanations for clarity.
Line 35
‘She had tree Caesarean Sections’ should be ‘She had three Caesarean Sections’.
References
Reference No. 4 appears to be a website and is not cited. I suggest omitting this reference.
Answers
Thank you for reviewing our manuscript and for your comments.
We have enlarged the pictures that you mentioned and we added extra notes about the details presented in the pictures.
We modified our spelling mistake from line 35.
Regarding the reference number 4, we deleted it.
Thank you very much,
Reviewer 2 Report
Comments and Suggestions for Authors
This is in interesting finding of a rare abdominal defect known as a "body stalk anomaly".
Although the images are fascinating, I think the readers would benefit from comparing a diagram with a drawn-out representation of the structures observed in the scans, or from labeling the structures seen on the scan images (eg: liver, limbs, stomach, celomic cavity).
Line 35: correct "tree" with "three"
Line 52: the authors mention "screening for chromosomal abnormalities"; was this a combined screening test or NIPT or other? Furthemore, although invasive testing was declined by the patient, was genetic testing carried out on the product of conception after termination? This may be useful for the patient's future pregnancies, although I aknowledge these abnormalities are not commonly linked to chromosomal abnormalities, as mentioned in line 30.
I would also suggest adding a brief literature review on abdominal wall defects to add information to the manuscript, and expanding the bibliography (in particular more recent literature regarding the effects of amnion disruption).
Author Response
This is in interesting finding of a rare abdominal defect known as a "body stalk anomaly".
Although the images are fascinating, I think the readers would benefit from comparing a diagram with a drawn-out representation of the structures observed in the scans, or from labeling the structures seen on the scan images (eg: liver, limbs, stomach, celomic cavity).
Line 35: correct "tree" with "three"
Line 52: the authors mention "screening for chromosomal abnormalities"; was this a combined screening test or NIPT or other? Furthemore, although invasive testing was declined by the patient, was genetic testing carried out on the product of conception after termination? This may be useful for the patient's future pregnancies, although I aknowledge these abnormalities are not commonly linked to chromosomal abnormalities, as mentioned in line 30.
I would also suggest adding a brief literature review on abdominal wall defects to add information to the manuscript, and expanding the bibliography (in particular more recent literature regarding the effects of amnion disruption).
Answers
Dear Reviewer,
Thank you for your comments on our manuscript. We added extra information on the pictures for a better understanding of the structures observed.
Regarding line 35 , we corrected the spelling mistake.
Line 52: We mentioned the screening for chromosomal abnormalities. We are sorry for the missed information. The screening was performed by a combination of ultrasound markers, such as nuchal translucency, nasal bone, ductus venosus, tricuspid valve flow and biochemistry (b-HCG and PAPP-A). She declined invasive testing and also further genetic investigations as she had already 3 children and did not want another pregnancy in the future. She opted for contraception.
As you advised us on adding more information about the abdominal wall defects and amnion rupture effects , we added extra bibliography in our manuscript.
Thank you very much,
Reviewer 3 Report
Comments and Suggestions for Authors
Dear authors,
this rare case report is not often mentioned in the literature. The description, including photographic documentation, is well done. I recommend to reconsider the article after minor revision.
Another comments
Lines 15-16
….., along with a short or absent umbilical cord.
Comment:
Body Stalk Anomaly causes abnormal development in the germinal disk, when the communication between the intraembryonic coelom (peritoneal cavity) and the extraembryonic coelom (chorionic cavity) remains open, so in most cases the umbilical cord does not develop and the liver with the venous duct grows directly into the placenta. A defect in the germinal disk between the epiblast and the hypoblast in the very early stages of embryogenesis is the basis of BSA. Some authors refer about absent umibilical cord. (e.g. Smrcek et al. Prenatal ultrasound diagnosis and management of body stalk anomaly: analysis of nine singleton and two multiple pregnancies. Ultrasound Obstet Gynecol 2003; 21: 322–328). I recommend slightly change the statement as follows: ….., along with a very short or absent umbilical cord.
General comments:
1. It would also be appropriate to add more information about this syndrome.
For example:
“Body stalk anomaly is a rare and severe malformation syndrome in which the exact pathophysiology and trigger factors are still unknown. Risk factors however are lower social status, drug abuse especially cocaine, maternal diabetes and maternal hemorrhagic disorder as well as stay at high altitude. Body stalk anomalies are generally not associated with chromosomal anomalies. Defects in genes related to embryogenesis may play a role.“
2. Since the article is submitted to the journal Diagnostics, it is appropriate to include a few sentences about diagnostic signs and differential diagnosis.
For example:
“This type of malformation might be responsible for a significant number of spontaneous abortions during the first trimester of pregnancy, and thus the real incidence for this anomaly might be underestimated. Ultrasound examination in the first trimester is major diagnostic tool. Ultrasound characteristics are presence of a major abdominal wall defect, severe kyphoscoliosis and absent or very short umbilical cord. Typically, the liver is directly attached to the placenta without interposed umbilical cord and there is major distortion of the spine. Exencephaly or encephalocoele, facial cleft, and limb amputations are common. Body stalk anomaly is accepted as a fatal anomaly, so it is important to differentiate it from other anterior wall defects (gastroschisis, omphalocele, bladder exstrophy, cloacal exstrophy, Cantrell pentalogy, and the OEIS complex - omphalocele, exstrophy of cloaca, imperforate anus and spinal defects) for evaluating the management options. The presence of the liver and intestine in the extraembryonic coelom differentiates body stalk anomalies from other subtypes.“
Or similarly.
Author Response
Dear authors,
this rare case report is not often mentioned in the literature. The description, including photographic documentation, is well done. I recommend to reconsider the article after minor revision.
Another comments
Lines 15-16
….., along with a short or absent umbilical cord.
Comment:
Body Stalk Anomaly causes abnormal development in the germinal disk, when the communication between the intraembryonic coelom (peritoneal cavity) and the extraembryonic coelom (chorionic cavity) remains open, so in most cases the umbilical cord does not develop and the liver with the venous duct grows directly into the placenta. A defect in the germinal disk between the epiblast and the hypoblast in the very early stages of embryogenesis is the basis of BSA. Some authors refer about absent umibilical cord. (e.g. Smrcek et al. Prenatal ultrasound diagnosis and management of body stalk anomaly: analysis of nine singleton and two multiple pregnancies. Ultrasound Obstet Gynecol 2003; 21: 322–328). I recommend slightly change the statement as follows: ….., along with a very short or absent umbilical cord.
General comments:
- It would also be appropriate to add more information about this syndrome.
For example:
“Body stalk anomaly is a rare and severe malformation syndrome in which the exact pathophysiology and trigger factors are still unknown. Risk factors however are lower social status, drug abuse especially cocaine, maternal diabetes and maternal hemorrhagic disorder as well as stay at high altitude. Body stalk anomalies are generally not associated with chromosomal anomalies. Defects in genes related to embryogenesis may play a role.“
- Since the article is submitted to the journal Diagnostics, it is appropriate to include a few sentences about diagnostic signs and differential diagnosis.
For example:
“This type of malformation might be responsible for a significant number of spontaneous abortions during the first trimester of pregnancy, and thus the real incidence for this anomaly might be underestimated. Ultrasound examination in the first trimester is major diagnostic tool. Ultrasound characteristics are presence of a major abdominal wall defect, severe kyphoscoliosis and absent or very short umbilical cord. Typically, the liver is directly attached to the placenta without interposed umbilical cord and there is major distortion of the spine. Exencephaly or encephalocoele, facial cleft, and limb amputations are common. Body stalk anomaly is accepted as a fatal anomaly, so it is important to differentiate it from other anterior wall defects (gastroschisis, omphalocele, bladder exstrophy, cloacal exstrophy, Cantrell pentalogy, and the OEIS complex - omphalocele, exstrophy of cloaca, imperforate anus and spinal defects) for evaluating the management options. The presence of the liver and intestine in the extraembryonic coelom differentiates body stalk anomalies from other subtypes.“
Or similarly.
Answers
Dear Reviewer,
We are thankful for your suggestions. Regarding the line 15 we modified it as you mentioned.
We added extra information about this anomaly, diagnostics signs and differential diagnosis as follows:
Body stalk anomaly (BSA) is a rare and serious syndrome characterized by severe malformations. Despite ongoing research, the exact cause and mechanisms behind this condition remain unclear. However, certain risk factors have been identified, such as lower socioeconomic status, maternal substance abuse (particularly cocaine), maternal diabetes and maternal hemorrhagic disorders [1]. It is important to note that body stalk anomalies are typically not linked to chromosomal abnormalities, although genetic defects related to embryonic development may contribute to their occurrence [1,2,3,4,5,6]. The prevalence of this condition is 1 in 14000 live births [2]. Body stalk anomaly is characterized by a large abdominal defect associated with spine and limb anomalies and a very short or absent umbilical cord [2]. There is no increased risk of recurrence in subsequent pregnancies. Body stalk anomaly is the most severe form and the rarest and is not compatible with life [3,5].
The ultrasound examination is an important diagnostic tool in the first trimester. The signs include abdominal wall defects, spine and limb abnormalities and more specifically abdominal contents herniated in the coelomic cavity, scoliosis and a very short umbilical cord [4].
A retrospective study of 17 cases of BSA diagnosed in a tertiary unit between 2009 and 2015 has proposed an algorithm for the differential diagnosis of abdominal wall defects in the first trimester [6]. This algorithm, based on specific diagnostic criteria, can aid in the accurate distinction between various abdominal wall defects, including BSA, during early pregnancy scans. Differential diagnosis of abdominal wall defects includes exomphalos, gastroschisis, cloaca extrophy and OEIS complex, pentalogy of Cantrell, abdominoschisis due to amniotic bands and BSA [6]. Abdominal organs attached to the placenta, severe kyphoscoliosis and absence of a free-floating umbilical cord is diagnostic for BSA. In exomphalos, the herniated viscera appear in the base of the umbilical cord, and a free-floating cord is visible in the amniotic cavity. In gastroschisis, Pentalogy of Cantrell, and cloaca extrophy, the eviscerated organs are within the amniotic cavity, and the umbilical cord is free-floating. In abdominoschisis due to amniotic bands, the amniotic membrane continuity is lost, but the umbilical cord is free-floating. In early amnion rupture, deformation and disruption of other structures, including craniofacial and limbs, can be demonstrated (limb-body wall complex). [6]
Thank you very much,
Round 2
Reviewer 1 Report
Comments and Suggestions for Authors
Thank you for providing the opportunity to review the revised paper titled 'Body Stalk Anomaly.' The authors responded to my suggestion appropriately, so I have no further concerns regarding this paper.
Author Response
Thank you for your efforts and support.
Reviewer 2 Report
Comments and Suggestions for Authors
issues have been addressed although more effort should be made to improve the background information on body stalk anomaly. If no further images of this case specifically are available, i would suggest adding diagrams for better comprehension.
Author Response
Thank you for your feedback. We have taken your suggestions into consideration and have added a diagram to the article to provide a visual aid for better comprehension of the condition. We believe that this diagram will enhance the understanding of body stalk anomaly and contribute to the overall quality of the article.
We appreciate your time and effort in reviewing our work, and we are confident that the changes made will address your concerns and improve the article's overall quality.
Sincerely,
Dr. Gica et al.